# Protective Effect of IgY Embedded in W/O/W Emulsion on LPS Enteritis-Induced Colonic Injury in Mice

**DOI:** 10.3390/nu16193361

**Published:** 2024-10-03

**Authors:** Zhaohui Wang, Ruihua Ye, Zijian Xu, Shidi Zhang, Chuanming Liu, Kongdi Zhu, Pengjie Wang, Jiaqiang Huang

**Affiliations:** 1Key Laboratory of Precision Nutrition and Food Quality, Department of Nutrition and Health, Ministry of Education, China Agricultural University, Beijing 100083, China; wzhaohui0222@163.com (Z.W.); xuzijian@cau.edu.cn (Z.X.); sy20233313708@cau.edu.cn (S.Z.); liucm0501@163.com (C.L.); zkd15938702692@163.com (K.Z.); 2College of Veterinary Medicine, China Agricultural University, Beijing 100083, China; yerh0522@cau.edu.cn; 3Beijing Advanced Innovation Center for Food Nutrition and Human Health, China Agricultural University, Beijing 100083, China

**Keywords:** enteritis, IgY, LPS, W/O/W

## Abstract

Chicken yolk immunoglobulin (IgY), an immunologically active component, is used as an alternative to antibiotics for the treatment of enteritis. In this study, IgY was embedded in a W/O/W emulsion to overcome the digestive barrier and to investigate the protective effect of IgY against LPS-induced enteritis in mice. Four different hydrophilic emulsifiers (T80, PC, SC, and WPI) were selected to prepare separate W/O/W emulsions for encapsulating IgY. The results showed that the IgY-embedded double emulsion in the WPI group was the most effective. IgY embedded in the W/O/W emulsion could reduce the damage of LPS to the mouse intestine and prevent LPS-induced intestinal mucosal damage in mice. It increased the number of cup cells, promoted the expression of Muc2, and increased the mRNA expression levels of *KLF3*, *TFF3*, *Itln1*, and *Ang4* (*p* < 0.05). It also enhanced the antioxidant capacity of the colon tissue, reduced the level of inflammatory factors in the colon tissue, and protected the integrity of the colon tissue. Stable embedding of IgY could be achieved using the W/O/W emulsion. In addition, the IgY-embedded W/O/W emulsion can be used as a dietary supplement to protect against LPS-induced enteritis in mice.

## 1. Introduction

In the 21st century, enteritis jeopardizes human and animal health worldwide, and in severe cases can even lead to death [1]. Gram-negative bacteria, such as Escherichia coli and Salmonella, cause intestinal bacterial infections [2]. Endotoxin is a component of the extracellular membrane of Gram-negative bacteria, which induces an inflammatory response and initiates the body’s immune response, causing damage to the intestinal mucosa [3,4]. Large amounts of intestinal bacteria and lipopolysaccharide (LPS) endotoxins invade the intestinal tissues and circulation, triggering bacterial transit and intestinal endotoxemia. Currently, immunomodulators such as antibiotics are used to treat LPS enteritis [5]. However, long-term use of these synthetic drugs not only leads to resistance but also causes severe immune side effects in animals [6]. Therefore, there is a need to find a drug with better efficacy that does not produce resistance and immune side effects for the treatment of LPS enteritis.

Chicken yolk immunoglobulin (IgY) is an efficient, inexpensive, and easily available immunoglobulin. IgY has a molecular weight of 180 kDa and its molecular structure is similar to that of IgG, a Y-type structure consisting of two heavy chains (H) and two light chains (L) with four constant regions (CH1-CH4) [7]. Extracting IgY antibodies from eggs is not only more productive, but also less harmful to the animals [8]. IgY treats gastrointestinal bacterial infections caused by *Salmonella* and *Helicobacter pylori* (*H. pylori*) [9,10]. Previous studies have shown that anti-lipopolysaccharide yolk antibody (anti-LPS IgY) is effective in neutralizing bacterial endotoxins in vivo, thereby reducing endotoxin damage [11]. Therefore, IgY can be used as an antibiotic alternative for the treatment of LPS-induced colitis. IgY can inhibit the activity and reduce the virulence of pathogenic bacteria by interacting with lipopolysaccharides and reducing or blocking their adherence and proliferation in the organism [12]. IgY exerts its immune response function through toxin neutralization and complement activation [7]. Thus, IgY antibodies are relevant alternatives for use as antimicrobial agents in the face of the emergence of drug-resistant bacteria in human and animal health. When administered orally, the antigen-binding activity of IgY is reduced or even completely lost due to hydrolysis by gastric acid and pepsin, thus greatly reducing its biological efficacy [13]. When IgY antibodies pass through the gastrointestinal tract, they are digested by the acidic gastric fluid environment [7]. In addition, the presence of biological barriers [14] can also lead to the decrease or even disappearance of the bioavailability of IgY antibodies, which is one of the current technical challenges in IgY research and application. W/O/W emulsions can be used in biologically active lipids and delivery systems to encapsulate, protect, and release hydrophilic components [12]. Water-in-oil-in-water (W/O/W) emulsions preserve and trap substances in the inner aqueous phase, as well as control the release of substances. These emulsions have been used as a means of microencapsulation for pharmacological (carriers of anticancer agents, hormones, steroids, etc.), cosmetic (creams containing encapsulated compounds are easy to apply), and other industrial applications [15].

Therefore, we used W/O/W emulsification to encapsulate IgY (1) to overcome the GI barrier, delivering IgY stably to the gut, and increase its bioavailability; (2) to investigate the effect of LPS-induced colonic injury and the protective effect of IgY embedded in W/O/W against colonic injury. Finally, we tested the effects of LPS-induced enteritis in mice by intraperitoneal injection of LPS and intervention with a W/O/W emulsion embedded with IgY.

## 2. Materials and Methods

### 2.1. Construction of IgY-Embedded Double Emulsion

An amount of 15 g [16] IgY (Unik, Beijing, China), NaCl (0.2% *w*/*w*), and 5 g sorbitol (Biotopped, Beijing, China) was added to the 30 g internal aqueous phase, (*w*), and the emulsifier PCPR (4% *w*/*w*) (Macklin, Shanghai, China) to the oil phase. Then, 40% (*w*/*w*) of the aqueous phase was added to 60% (*w*/*w*) of the oil phase and the mixture was mixed at 11,000 rpm for 3 min using a high-shear mixer (T25 digital ULTRA-TURRAX, IKA, Shanghai, China) to produce a primary emulsion. The primary emulsion was then homogenized at a pressure of 60 MPa (NS 1001, GEA Niro Soavi, Parma, Italy) to produce the final W/O emulsion.

Tween-80 (T80) (4% *w*/*w*) (Macklin, Shanghai, China), lecithin (PC) (4% *w*/*w*) (BZ52097, Biotopped, Beijing, China), sodium caseinate (SC) (4% *w*/*w*) (Biotopped, Beijing, China), and whey protein powder (WPI) (4% *w*/*w*) (Biotopped, Beijing, China) was mixed as hydrophilic emulsifiers to form the outer aqueous phase. Next, 40% of the primary W/O emulsion dispersed in 60% of the external aqueous phase was sheared for 3 min at 10,000 rpm using a high-shear mixer (T25 digital ULTRA-TURRAX, IKA, Shanghai, China) to produce the final W/O/W emulsion.

### 2.2. Stability Analysis of Double Emulsion Embedding

A Mastersizer 2000 (Malvern Instruments, Malvern, UK) was used to determine the droplet size distribution of the W/O/W emulsions. Samples were dispersed in distilled water (refractive index 1.33) and the rotational velocity was kept at 1750 rpm. Measurements were performed after the preparation of the double emulsions.

The microstructure of the fresh emulsions was observed with an optical microscope (Axio Scope.A1, Carl Zeiss, Jena, Germany) with a 100× oil immersion objective lens. Images were captured with AxioVision Rel. 4.8 software (Carl Zeiss, Jena, Germany).

The kinetic stability of the double emulsions was monitored with the optical scanning instrument Turbiscan ASG (Formulation, Poitiers, France). Immediately after preparation, the emulsions were placed in flat-bottomed cylindrical glass tubes (140 mm height, 16 mm diameter) and the first measurement of backscattered light intensity (day 0) was performed. Tubes were stored at 25 °C and subsequent measurements were taken on day 1 at a wavelength of 880 nm. Backscattering (BS) profiles at different sample heights (mm) were used to analyze the destabilization of emulsions. A sample height of 0 mm corresponds to the bottom of the measuring cell. The Turbiscan Stability Index (TSI), calculated as the sum of all destabilization processes in the sample along the measurement cell, was quantified according to Equation (1).
(1)$$\sum_{j}\left|scanref\left(hj\right)-scani\left(hi\right)\right|=TSI$$
where scan_ref_ and scan_i_ are the initial backscattering value and the backscattering value after 1 day of storage, respectively, hj is the given height in the measuring cell, and TSI is the sum of all the scan differences from the bottom to the top of the tube.

### 2.3. In Vitro Evaluation of IgY Embedded in W/O/W

#### 2.3.1. Tolerance Test against Artificial Gastric Juice

In order to determine whether the W/O/W emulsion with embedded IgY could overcome the digestive obstacles of the gastric environment, an in vitro simulated gastric fluid test was performed. Unencapsulated IgY and IgY encapsulated with four different double emulsions were added to the simulated gastric fluid and the IgY activity was detected by an Elisa kit (Jiangsu Jingmei Biotechnology Co., Ltd., Jiangsu, China) after 30, 60, 90, and 120 min.

#### 2.3.2. Tolerance Test against Artificial Intestine Juice

In order to determine whether the W/O/W emulsion encapsulating IgY can be digested by the intestine and release IgY, an in vitro simulated intestinal fluid test was performed. Unencapsulated IgY and IgY encapsulated with four different double emulsions were added to the simulated intestinal fluid and the IgY activity was detected by an Elisa kit (Jiangsu Jingmei Biotechnology Co., Ltd., Yancheng, China) after 60, 120, 240, 720, and 960 min, respectively.

### 2.4. Animals

All animal experiments were approved by the Animal Care and Use Committee of the China Agricultural University under license number AW32504202-5-5; the approval date is 23 May 2024. A total of 50 male healthy KM mice were purchased from Spectrum USA (Beijing, China). The 50 mice were randomly divided into five groups: control group (Control; *n =* 10), LPS-induced (5 mg/kg) group (LPS; *n =* 10), group receiving IgY treatment and undergoing LPS induction (IgY; *n* = 10), receiving double emulsion treatment and undergoing LPS induction group (DE; *n* = 10), and receiving embedded IgY double emulsion treatment and undergoing LPS induction group (IgY + DE; *n* = 10). Briefly, mice in each group were gavaged continuously for 14 d. The control and LPS groups were gavaged with 0.5 mg of 0.9% NaCl, the IgY group was gavaged with 0.5 mg of unembedded IgY, the DE group was gavaged with 0.5 mg of double emulsion, and the IgY + DE group was gavaged with 0.5 mg of double emulsion with embedded IgY. In the control group, 0.9% NaCl was injected, and the other four groups were injected intraperitoneally with 5 mg/kg LPS.

At the end of the experiment, the 50 mice were euthanized by cervical dislocation. After removal of the colon tissue, it was fixed in 10% formalin for one week for subsequent histological analysis. The remaining colon tissues were stored frozen at −80 °C.

### 2.5. Histological Analysis

Paraffin-embedded colon tissue was sectioned. Specimens were sectioned and stained with hematoxylin and eosin (H & E). Measurements of villus height and crypt depth were performed using Image J (version 1.48r, National Institutes of Health, Bethesda, MD, USA). The sections were subjected to periodic acid Schiff and Alcian Blue (PAS/AB) staining and the number of goblet cells was counted using Image J (National Institutes of Health, Bethesda, MD, USA). The blue area and total area were measured separately in the colon. The proportion of goblet cells was quantified based on the ratio between the blue area and the total area.

### 2.6. Immunohistochemical

Sections were subjected to antigenic microwave repair, closed with 5% goat serum, and incubated overnight at 4 °C with MUC 2 antibody (1/2000, Abcam Co., Inc., Cambridge, UK). Then, the sections were washed with PBS (pH 7.0), and sheep anti-rabbit IgG (CoWin Biotech Co., Inc., Beijing, China) was added dropwise for 2 h. Then, the color was developed using a diaminobenzidine (DAB) kit (Zhongshan Jinqiao Biotech Co., Ltd., Beijing, China) and the samples were restained with hematoxylin for 5 min. Positive cells were counted in 25 randomized areas of 5 intestinal cross-sections per sample. The mean IOD of positive cells was measured using ImageJ (National Institutes of Health, Bethesda, MD, USA).

### 2.7. ELISA Detected TNF-α, IL-1β, IL-6, and IL-12

Take 100 mg of colon tissue from −80 °C and place it in a 2 mL centrifuge tube add 900 μL pre-cooled PBS, marked and homogenized using a tissue homogenizer. After homogenization, centrifuge 5000 g for 10 min and take the supernatant for later use. Use the Elisa reagent kit (Jiangsu Jingmei Biotechnology Co., Ltd., Yancheng, China) for testing.

### 2.8. Detection of Redox Indicators

An amount of 100 mg of −80 °C colon tissue was added into a 2 mL centrifuge tube, and 900 µL of pre-cooled PBS was added, labeled, and homogenized with a tissue homogenizer. After homogenization, centrifugation was carried out at 5000× g″ for 10 min and the supernatant was removed. Catalase (CAT), malondialdehyde (MDA), glutathione peroxidase (GSH Px), and superoxide dismutase (SOD) were detected using specific kits (Nanjing Jiancheng, Nanjing, China).

### 2.9. RT-qPCR

Total RNA from colon tissue was extracted with TRIzol reagent (CW0580; Coyne Biotech, Beijing, China), and the samples and TRIzol were loaded into RNase-free centrifuge tubes and mixed by shaking. They were allowed to stand at 4 °C for 5 min, chloroform was added, and then they were allowed to stand. The samples were centrifuged and the supernatant removed. The same volume of isopropanol was added and mixed well. The samples were centrifuged and the supernatant discarded. To the precipitate, 75% ethanol was added and mixed, and then the sample was centrifuged and the supernatant discarded. After drying the precipitate, RNA was solubilized by adding RNase-free water and RNA concentration and purity were measured with a NanoPhotometer (P330; Implen, Munich, Germany). cDNA was prepared by reverse transcription of 1 μg RNA for quantitative qPCR (HiScript QRTsupermix (R312-02; Vazyme Biotech, Nanjing, China)). Real-time qPCR was performed using SYBR Green Master Mix (Q141-02; Vazyme Biotech, Nanjing, China) and a OneStep Plus instrument (Applied Biosystems, Inc. Waltham, MA, USA). Primers are shown in Table 1.

### 2.10. Stastical Analysis

Data were analyzed and plotted using GraphPad Prism 10.0 software and results are expressed as “mean ± standard deviation”. The results are expressed as mean ± standard error of mean (SEM) of independent experiments. Data were analyzed by one-way ANOVA: * *p* < 0.05, ** *p* < 0.01, *** *p* < 0.001, **** *p* < 0.0001.

## 3. Results

### 3.1. Characterize of IgY W/O/W Emulsion

The preparation process of the W/O/W emulsions is shown in (Figure 1a). Firstly, in the preparation of the water-in-oil emulsions, we chose different concentration gradients of the lipophilic emulsifier polyglyceryl trinitrate (PGPR), and the results showed that emulsion delamination occurred in 1%, 2%, and 3% concentrations of PGPR, while the 4% concentration of PGPR formed stable water-in-oil emulsions (Figure 1b). Next, while preparing the W/O/W emulsions, we found that the T80 and PC groups experienced emulsion breakage and the emulsions showed severe water–oil delamination (Figure 1c), but this did not occur in the SC and WPI groups. Microscopic analysis also revealed emulsion breakage in the T80 and PC groups (Figure 1d). Subsequently, we measured the droplet size and found two peaks of different sizes in both the T80 and PC groups, again indicating that the emulsion droplets were of different sizes, whereas the droplet sizes in the SC and WPI groups were uniform (Figure 1e).

### 3.2. Stability Analysis of W/O/W Emulsions Prepared with Different Emulsifiers

Analysis of the backscattering spectrograms revealed that the WPI emulsion had the best stability during storage at 25 °C, while the other three emulsions showed varying degrees of upwelling aggregation and sinking (Figure 2a). After the measurement of emulsion kinetic stability, we also found that the WPI group emulsions had the highest stability, with no uplift or delamination of the emulsions (Figure 2b).

### 3.3. Stability Analysis and In Vitro Validation of Different W/O/W Emulsions

Next, we analyzed the TSI values of the different W/O/W emulsions. The Turbiscan stability index (TSI), calculated as the sum of all destabilization processes of the sample along the measurement cell, was quantified according to Equation (1). We found that the W/O/W emulsions of the WPI group had the smallest change in TSI values, indicating that these emulsions were the most stable (Figure 3a). The simulated gastric fluid results revealed that the WPI group emulsions had the highest encapsulation rate of 80.2%, while the rest of the emulsions had encapsulation rates of 20% and below (*p* < 0.0001) (Figure 3b). In simulated intestinal fluid, the WPI group emulsions had a good release rate (*p* < 0.0001) (Figure 3c). This proved that the W/O/W emulsions could be digested by the intestines while remaining intact in the gastric environment, indicating that the WPI-type W/O/W emulsions could achieve stable delivery of IgY to the intestines.

### 3.4. IgY + DE Ameliorates LPS-Induced Intestinal Damage

Next, we validated LPS-induced colonic inflammation in mice using an IgY-embedded W/O/W emulsion (hereafter referred to as IgY + DE). Compared with the control group, LPS induction performed after 14 consecutive days of gavage treatment in 10 mice (Figure 4a) led to weight loss (Figure 4b), which was significantly alleviated after intervention with IgY + DE. The intestinal length of the mice was measured and counted after execution (Figure 4c,d), and it was found that the LPS-induced intestinal shortening in mice was ameliorated after IgY + DE intervention (*p* < 0.001). Based on HE-stained sections and histological scores, we found that after LPS induction in mice colon tissues compared to the control group, the colon tissues in the LPS group were significantly damaged, as evidenced by severe infiltration of inflammatory factors, villous epithelial atrophy, and crypt epithelial detachment, which were attenuated by the intervention of IgY + DE (*p* < 0.05) (Figure 4e,f).

### 3.5. IgY + DE Alleviates LPS-Induced Colitis by Improving the Mucosal Barrier

Next, we found that the number of goblet cells was significantly reduced in the colonic tissues of LPS-treated mice compared with the control group by AB-PAS staining and goblet cell counting, which was significantly alleviated by IgY + DE intervention (*p* < 0.0001) (Figure 5a,b). Muc2 is secreted mainly by goblet cells and enters the intestinal lumen to form a mucus layer. We examined the expression of Muc2 in colonic tissues by immunohistochemistry. We found that LPS induction resulted in downregulation of Muc2 expression compared to the control group, which was ameliorated in the IgY + DE group (*p* < 0.0001) (Figure 5b,c). The transcription factor *Klf3* maintains intestinal barrier function, and *Tff3* protects thrush cells and promotes mucosal repair. According to the results, we found that the mRNA levels of *Tff 3* (*p* < 0.0001) (Figure 5e), *Klf3* (*p* < 0.01) (Figure 5f), and *Muc2* (*p* < 0.001) (Figure 5g) were more severely reduced in the LPS group than in the control group, which was alleviated after IgY + DE intervention. We also examined the expression of genes related to mucosal barrier repair in colon tissues, and the results revealed that *Retnlb* (*p* < 0.001) (Figure 5h), *Itln1* (*p* < 0.01) (Figure 5i), and *Ang4* (*p* < 0.0001) (Figure 5j) expressions were significantly downregulated in colon tissues compared to the control group, and this was mitigated in the IgY + DE group.

### 3.6. IgY + DE Alleviates LPS-Induced Oxidative Stress in the Colon

We further investigated the effects of LPS treatment on intestinal antioxidant capacity (CAT, MAD, SOD, and GSH-Px). Consistent with our predictions, the levels of CAT (*p* < 0.01), SOD (*p* < 0.05), and GSH-Px (*p* < 0.01) were significantly decreased in the LPS group compared to the control group, whereas no significant changes were observed in the IgY + DE group (Figure 6a,b,d). However, MDA levels were significantly higher in the LPS group compared to the control group, while no significant changes were observed in the IgY + DE group (*p* < 0.01) (Figure 6c). These results suggest that IgY + DE treatment can alleviate the weakened antioxidant capacity of the colon induced by LPS.

### 3.7. IgY + DE Reduces the Colonic Inflammatory Response Caused by LPS

Next, we examined the levels of inflammatory factors in the colonic tissues of mice. Based on the results, we found that the levels of pro-inflammatory cytokines IL-6 (*p* < 0.0001) and TNF-a (*p* < 0.05) were significantly elevated in the LPS group compared to the control group, whereas no significant changes were observed in IgY + DE (Figure 7a,c). The expression of anti-inflammatory cytokines IL-10 (*p* < 0.05) and IL-1β (*p* < 0.01) was significantly decreased in the LPS group compared to the control group, which was alleviated by IgY + DE (Figure 7b,d).

## 4. Discussion

Four food-grade emulsifiers (T80, PC, SC, and WPI) were tested and analyzed, and water-soluble emulsifiers suitable for IgY embedding with good stability were selected. In addition, there was some compatibility between high-HLB-type emulsifiers and low-HLB-type emulsifiers. On this basis, the oil–water interface and water–oil interface were minimized to reduce or eliminate the interfacial instability and emulsification instability. Four different concentrations of PGPR were selected for the experiment to prepare W/O emulsions, and it was found that a 4% concentration of PGPR was the most stable (Figure 1b). This may be due to the fact that increasing the concentration of the emulsifier can reduce the surface tension at the oil–water interface and play a stabilizing role in water-in-water emulsion. Small-molecule emulsifiers T80 and PC, the protein emulsifier SC, and the natural emulsifier WPI were selected and characterized. Here, PC is an analytical standard-grade soy lecithin. To prepare stable double emulsion-embedded IgY as a dietary supplement, natural macromolecule WPI was used as an emulsifier. Early studies have found that emulsifiers with large molecular sizes (WPI) do not diffuse between interfaces, which improves the stability of W/O/W emulsions and contributes to the maintenance of double emulsion stability [17,18,19]. Considering food applications, natural emulsifiers such as proteins (WPI) have the advantage of being biodegradable, biocompatible, and non-toxic [20,21]. It has been reported that the stability of W/O/W double emulsions can be improved when low concentrations of WPI are added to the internal water phase [22,23]. According to the microscopic results, the W/O/W emulsions of the WPI and SC groups were the most stable, and no emulsion breakage was observed (Figure 1c,d). Microscopy of double emulsions prepared with PC and T80 reveals many fine particles (Figure 1c,d), which is mainly due to the fact that the adsorption energies of the small-molecule emulsifiers are similar to the thermodynamic energy (less than 10 times the thermodynamic energy). That is, the small-molecule emulsifiers are thermally migratory at the interface, which means that the thermally migratory motion of these emulsifiers at the interface provides them with enough energy to desorb from the interface. Once the emulsifiers are desorbed, this inevitably leads to a decrease in the stability of the entire double emulsion [24]. Among the particle sizes of the double emulsions made with the four different emulsifiers, the lecithin group had a smaller particle size than the T80, sodium caseinate, and WPI groups, which may be related to the emulsifying properties of the emulsifiers (Figure 1e). Compared to protein-based emulsifiers, small-molecule emulsifiers can be adsorbed and dispersed rapidly at the oil–water interface, which makes it easier to form emulsions [25]. The double emulsions produced by WPI have a uniform particle size, indicating that WPI has a higher encapsulation rate than either of the other two groups and does not result in suspension and sedimentation or emulsion breakage.

After analyzing the stability of W/O/W emulsions, it was observed that the WPI group had the best emulsion stability (Figure 2a,b and Figure 3a) The stability index (TSI) of the specimen system was negatively correlated with the stability of the emulsion [26]. The variation in TSI in double emulsions is mainly dependent on factors such as emulsion particle size (polymerization/flocculation) and droplet migration [27]. Since small-molecule emulsifiers usually consist of low-molecular-weight droplets with a small spacing between two droplets, they can easily approach each other and form agglomerates [27]. However, the particle size of particulate stabilizers is usually greater than 10 nm, which is much larger than that of small- or large-molecule emulsifiers. After the particles enter the oil–water interface, it is difficult to form a tight bond between two neighboring interfaces due to the large particle size. Therefore, this is very beneficial to the stability of the emulsion.

However, although the WPI group double emulsions showed stronger encapsulation stability among the four groups of emulsions, they still had the limitation of being difficult to store for a long period due to the restriction of their double emulsion structure. In this study, we found that the T80, SC, and PC double emulsions showed different degrees of water–oil delamination on the 3rd day under the storage condition of 4 °C. Even the best stabilized WPI group double emulsion showed significant water–oil delamination on day 5. The same results were shown in a previous study, where after storing the double emulsions for 7 days, the presence of a clear interface between the groups of double emulsions (serum and cream layers) highlighted the unstable process of emulsification in all systems as well as the accumulation of the WPI system [20]. This may be due to the inability of double emulsions to achieve a balance between stability and controlled release properties during long-term storage [22]. Therefore, when formulating emulsions containing additives, emphasis should be placed on the balance between stability and controlled release properties. We can control the encapsulation and release efficiency of double emulsion microcapsules by controlling the shear strength during the construction of W/O/W emulsions [28]. It has been shown that the balance between stability and the controlled release properties of W/O/W emulsions can be achieved by employing a defined amount of the lipophilic emulsifier isopropyl myristate (IPM), which has the advantages of a high encapsulation rate, high centrifugal retention, uniform particle size distribution, and exhibits stability over 6 months [29]. In addition, alendronate microspheres prepared by the W/O1/O2 emulsification technique by adjusting the ratio of aqueous and oil phases had maximum drug loading and good overall performance in vitro, and are expected to be used as a bisphosphonate drug delivery system for dental and other clinical applications [30]. It has been shown that the addition of a small amount of non-toxic mustard oil [31] or polymer-encapsulated gaseous chlorine dioxide (ClO_2_) [32] in the construction of W/O/W emulsions can reach the goal of inhibiting the growth of bacteria and prolonging the stability of W/O/W emulsions.

Encapsulated IgY double emulsions as dietary supplements need to pass through gastric juice before reaching the intestinal tract in order to be released and functional. The results showed that unencapsulated IgY was inactivated immediately upon entering the simulated gastric fluid. Consistent with previous reports, the activity of unencapsulated IgY in the simulated gastric fluid was reduced by 80–90% or even completely inactivated [16]. Lee et al. reported that the activity of IgY was stable under neutral and alkaline conditions but was reduced by 80% in low-acidic environments [33]. Pepsin and hydrolyzed proteins were also present in the gastric juice and the presence of pepsin also decreased the activity of IgY. The same trend was observed in the T80 group, which suggests that the T80 double emulsion is not encapsulation-stable. The WPI group of IgY showed the highest survival rate in the simulated gastric fluid, which could reach 80.2% (Figure 3b). The protective effect of the double emulsion prepared with WPI as the hydrophilic emulsifier on IgY was confirmed, suggesting that encapsulation by the double emulsion could protect IgY from inactivation in the gastric fluid environment in the presence of low acid and pepsin. In simulated intestinal fluid, the results showed that four different double emulsions displayed different IgY release rates after passing through simulated intestinal fluid. Among them, the T80, PC and SC groups released almost all the IgY in the simulated intestinal fluid within 1 h. The stability of the double emulsions remained unchanged after 2 h of entry into the simulated gastric fluid (Figure 3c). After entering the intestinal digestive fluid, the stability of the emulsion began to decrease, which may be attributed to the proteases and lipases in the pancreatic enzymes digesting the proteins adsorbed at the interface of the droplets, disrupting the interfacial adsorbent membrane, leading to droplet coalescence and a decrease in the stability of the double emulsion [34]. Consistent with previous results, the stability of the double emulsion remained essentially unchanged in the simulated oral and gastric fluid environments but decreased significantly upon entry into the simulated intestinal fluid [34]. This suggests that the double emulsion is degraded by digestion in the intestinal fluid and IgY is released from the internal aqueous phase into the gut [16]. It was found that some natural macromolecules (WPI and WPH) can form a thick and strong gel-like internal interface inside the double emulsion, and this structure is favorable for controlled release [22,35]. The experimental results showed that the double emulsion of the WPI group was an effective material to protect IgY from the gastric fluid environment and that the material stabilized the release of IgY in the intestine.

In past studies, due to ease of manipulation as well as reproducibility of inflammation induction, strong lipopolysaccharide-induced mouse models of inflammation have been widely used, such as the induction of intestinal inflammation and hepatic inflammation [24,26,36]. As an inflammatory mediator, LPS stimulates the systemic immune response, leading to inflammation production, disrupting intestinal mucosal structure and function, and inducing secretion of a range of inflammatory factors [27]. The use of intraperitoneal LPS to induce intestinal injury and inflammation is a widely accepted model for inducing acute intestinal inflammation. This model helps to identify functional substances that are effective in preventing or reducing acute intestinal inflammation. Intestinal diseases caused by chronic inflammation are reported to be one of the most important threats to human health. In addition to antibiotic treatment, some trace minerals and their proteins such as selenium and SELENOI have been shown to be effective in alleviating intestinal disorders [37,38,39,40]. In recent years, IgY has attracted attention for its ability to inhibit intestinal inflammation, improve barrier function, and optimize intestinal structure [41]. IgY can be used as an antibiotic replacement drug for LPS-induced enterocolitis. IgY exerts its immunoactive function through toxin neutralization and complement activation [42]. In this study, we found that IgY could maintain intestinal health by reducing LPS-induced colonic injury, enhancing the intestinal mucosal barrier, reducing inflammatory responses in colonic tissues, and enhancing their antioxidant capacity.

In this study, 5 mg/kg of LPS significantly reduced the body weight of mice. IgY that had been embedded in the W/O/W emulsion attenuated the lipopolysaccharide-induced weight loss (Figure 4b). In addition, histological scores showed that mice in the LPS group had significant colonic tissue damage, as evidenced by shortening of intestinal tissues, compared with the control group (Figure 4c,d). IgY was able to reduce the shortening of colon length in mice caused by LPS, suggesting that IgY helps to resist the intestinal damage caused by endotoxin in mice. IgY supplementation improved the structure and barrier function of the colon and reduced intestinal inflammation after LPS induction (Figure 4e,f). Consistent with previous studies [34,43,44], LPS-induced damage to intestinal villi caused histological damage and hindered intestinal development. In addition, disruption of intestinal morphology and structure leads to impaired digestion and absorption in the body. In the present study, we found that continuous IgY supplementation for 14 days attenuated histological damage in the colon after lipopolysaccharide attack, which is similar to the results reported by Li et al. [45]

It has been reported that mucus composed of mucin coats the intestinal epithelial cells, thereby protecting the intestinal mucosa from invasion by pathogens. According to Alcian Blue–Periodic Acid–Schiff (AB-PAS) staining, thrush cells were dark blue in color and distributed in the intestinal epithelial cells. In our study, the number of thrush cells was significantly higher in the IgY + DE group than in the LPS group (Figure 5a,b). In addition, the most abundant mucin in the mucus layer was Muc2 [46,47,48]. Notably, LPS stimulation led to a decrease in Muc2 expression in the mucus layer (Figure 5b,c). Muc2 is mainly secreted by goblet cells, and this decrease is therefore due to a decrease in the number of goblet cells and a downregulation of Muc2 expression. The transcription factor *Klf3* is involved in protecting the intestinal barrier, and *Tff3* protects cup cells and promotes mucosal repair. Consistent with our prediction, the expression levels of *Muc2* and *Klf3* mRNA and *Tff3* mRNA were higher in the IgY + DE group than in the LPS group (Figure 5e–g). This suggests that IgY intervention can significantly increase the expression of Muc2 in the colon, enhance the mucus layer barrier, and maintain intestinal health. These results suggest that IgY supplementation improves the morphological development of the colon and maintains the integrity of the colon tissue and the intestinal mucosal barrier function by enhancing the expression of Muc2. We also examined the expression of genes related to mucosal barrier repair in colon tissues, and the results revealed that *Retnlb* (Figure 4e), *Itln1* (Figure 4e), and *Ang4* (Figure 4e) expression was significantly downregulated in colon tissues compared to the control group, and this was mitigated in the IgY + DE group.

We have further investigated the markers of oxidative stress in colon tissues. Malondialdehyde may cause cytotoxicity, which is a byproduct of lipid peroxidation [49]. High intracellular levels of GSH and CAT catalyze the production of O_2_ and H_2_O_2_ by superoxide anion radicals, which protects cellular structure and function from oxidative damage [50]. Consistent with our prediction, GSH-Px, SOD values and CAT levels were significantly reduced in the LPS group. After IgY supplementation, the levels of GSH-Px, SOD, and CAT were increased. However, malondialdehyde (a lipid peroxidation end product) levels were significantly higher in the IgY + DE group compared to the LPS group. (Figure 7a–d). The above results indicated that LPS treatment weakened the antioxidant capacity of the colon, whereas IgY supplementation enhanced the antioxidant capacity of the colon tissue.

Inflammation is an important pathway involved in the immune regulation of the body. Notably, LPS-induced intestinal inflammation was attenuated by IgY + DE intervention. LPS induction leads to impaired morphology of colonic tissues, disruption of the intestinal mucosal barrier, and the development of a strong inflammatory response in mice. In addition, LPS-induced inflammation in colonic tissues resulted in increased secretion of inflammatory cytokines (e.g., IL-1β, IL-6, and TNF-α) [51], which further suggests that IgY + DE intervention can exert anti-inflammatory effects by downregulating the expression of inflammatory factors.

Existing commercially available formulations with efficacy against colonic injury caused by enteritis are available as fluoroquinolones. Fluoroquinolones are combinations of isomeric compounds used in hospitals and the population for the treatment of a wide range of serious infections [52]. Fluoroquinolone antibiotics have become the mainstay of antimicrobials and are effective against colitis, mycobacterial infections, respiratory tract infections, urinary tract infections, and skin infections [52]. Fluoroquinolones have the same antimicrobial and anti-inflammatory properties as IgY, but post-marketing surveillance has shown phototoxicity, prolonged QT interval, and anaphylaxis as side effects [53]. And, fluoroquinolones are restricted in the pediatric population [54,55]. In contrast, IgY antibodies are safe, an immunologically active ingredient located in poultry eggs from a safe and healthy source [7]. False positives do not occur in immunoassays due to phylogenetic disorders. IgY itself does not cause immune resistance in the body [56,57]. These differences provide significant advantages for IgY in many areas of research (e.g., antibiotic replacement therapy) [58].

## 5. Conclusions

These results indicate that a W/O/W emulsion can achieve stable embedding of IgY and improve the bioavailability of IgY. And it has a protective effect as a dietary supplement against LPS-induced colitis in mice. In addition, IgY can be regarded as a beneficial food for regulating intestinal health, which provides a basis for further understanding of the beneficial effects of IgY.

## Figures and Tables

**Figure 1 nutrients-16-03361-f001:**
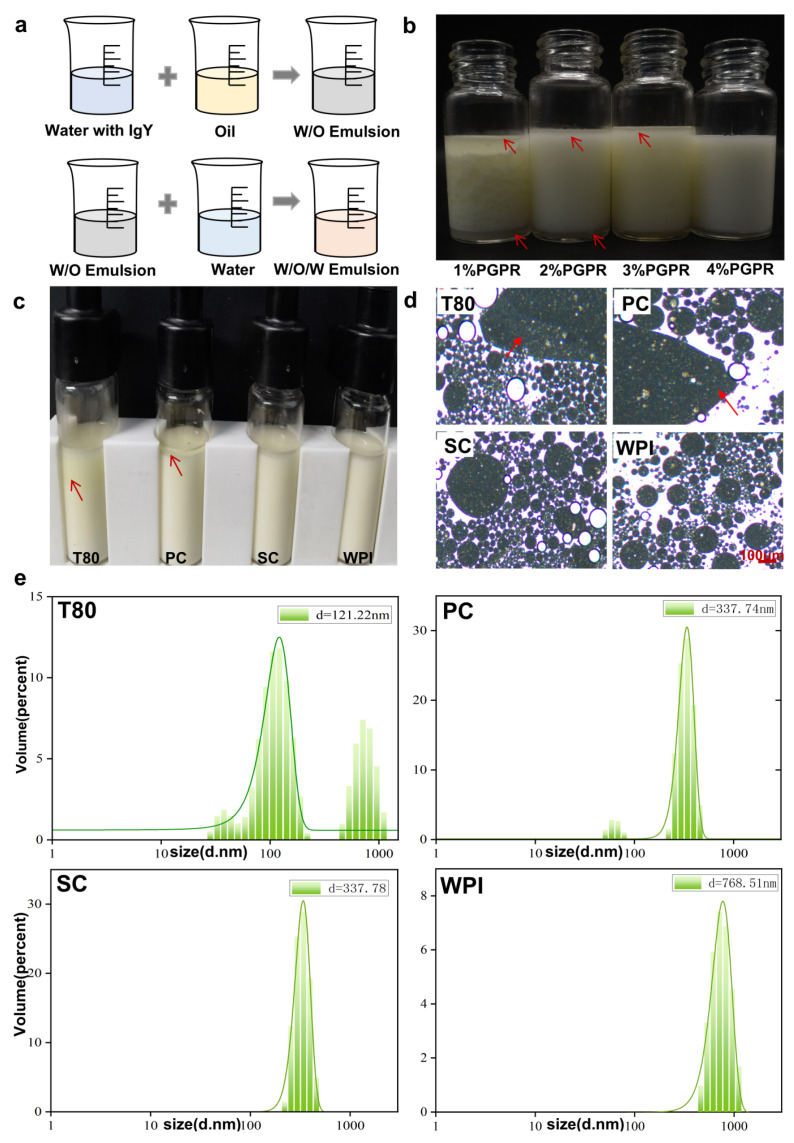
Characterization of embedded IgYW/O/W emulsions (**a**) Preparation process of W/O/W emulsion. (**b**) Observations on the stability of W/O emulsions with different PGPR concentrations. (Red arrows indicate emulsion layering.) (**c**) Observation on the stability of W/O/W emulsions prepared with different emulsifiers. (Red arrows indicate emulsion layering.) (**d**) Microscopic observation of W/O/W emulsions prepared with different emulsifiers. (Red arrows show breaking of the emulsion.) (**e**) Measurement of particle size of W/O/W emulsions prepared with different emulsifiers. T80, Tween-80 group IgY double emulsion; PC, lecithin group IgY double emulsion; SC, sodium caseinate group IgY double emulsion; WPI, whey protein powder group IgY double emulsion.

**Figure 2 nutrients-16-03361-f002:**
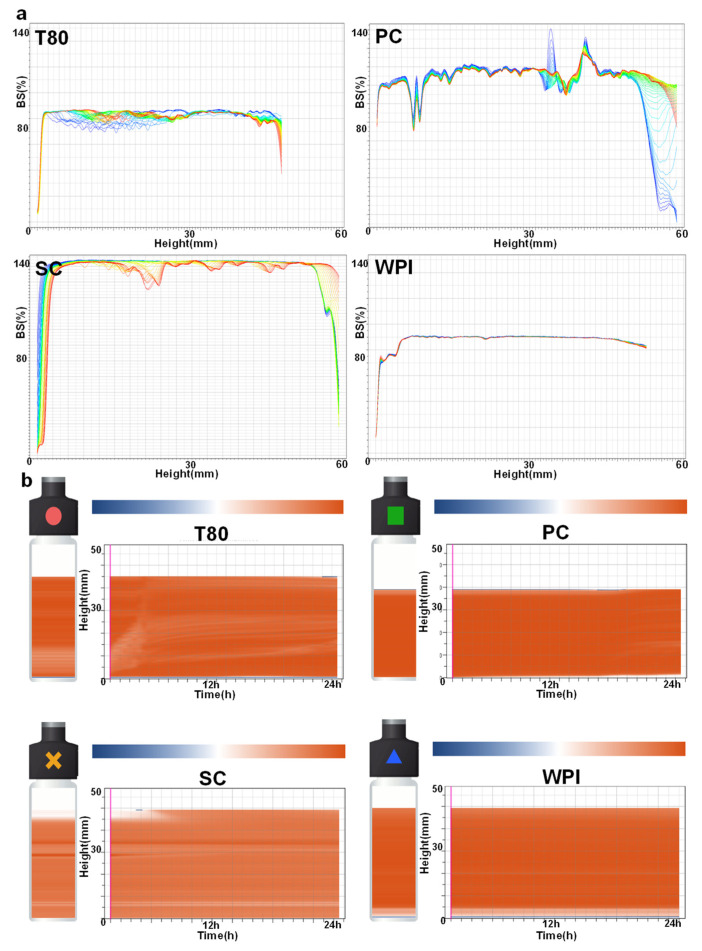
Stability analysis of W/O/W emulsions prepared with different emulsifiers (**a**) Backscattering spectra of different W/O/W emulsions. (**b**) Kinetic stability analysis of different W/O/W emulsions. T80, Tween-80 group IgY double emulsion; PC, lecithin group IgY double emulsion; SC, sodium caseinate group IgY double emulsion; WPI, whey protein powder group IgY double emulsion.

**Figure 3 nutrients-16-03361-f003:**
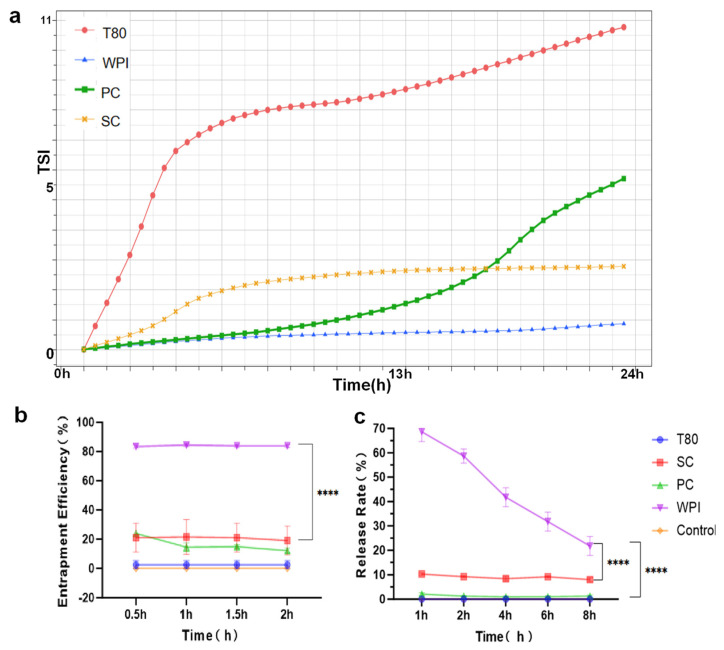
Stability analysis and in vitro validation of different W/O/W emulsions. (**a**) Stability analysis of different W/O/W emulsions. (**b**) In vitro-simulated stomach liquid (*n* = 5). (**c**) In vitro-simulated intestinal fluid (*n* = 5). Data were analyzed by one-way ANOVA: **** *p* < 0.0001. T80, Tween-80 group IgY double emulsion; PC, lecithin group IgY double emulsion; SC, sodium caseinate group IgY double emulsion; WPI, whey protein powder group IgY double emulsion.

**Figure 4 nutrients-16-03361-f004:**
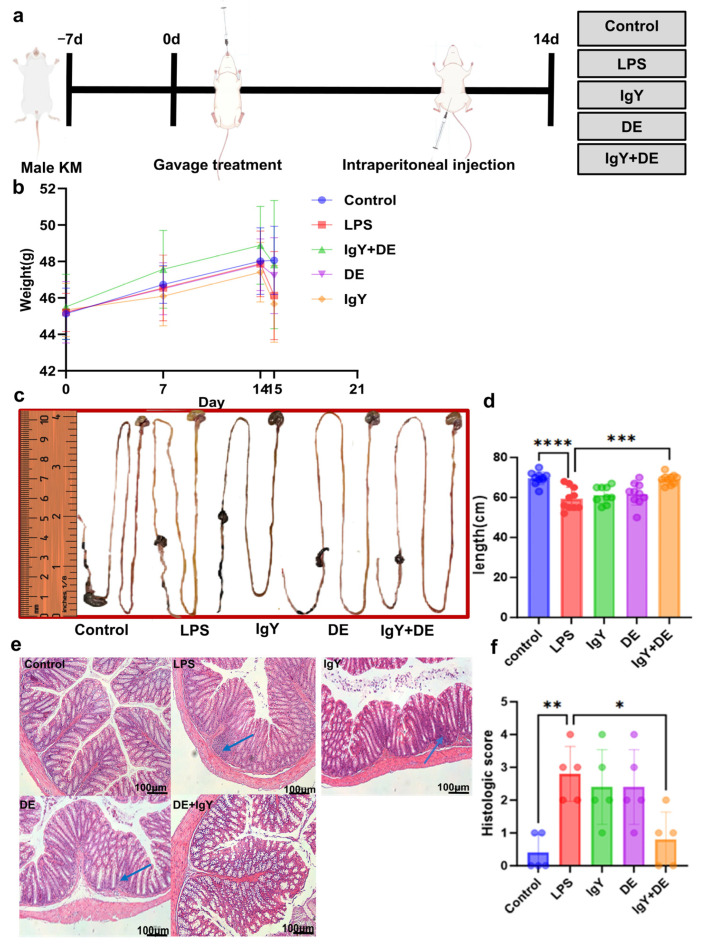
IgY + DE ameliorates LPS-induced intestinal damage. (**a**) Experimental research design. (**b**) Mouse body weight analysis (*n* = 10). (**c**) Histologic scores. (**d**) Mouse intestinal comparison. (**e**) Intestinal length analysis in mice (*n* = 10). (**f**) Colon histopathology (H & E), 100 μm. (Blue arrows indicate inflammatory factor infiltration.) Data were analyzed by one-way ANOVA followed by Tukey’s post hoc test: * *p* < 0.05, ** *p* < 0.01, *** *p* < 0.001, **** *p* < 0.0001. Control: physiological saline gavage + intraperitoneal injection of physiological saline; LPS: physiological saline gavage + LPS intraperitoneal injection; IgY: IgY gavage + LPS intraperitoneal injection; DE: double emulsion gavage + LPS intraperitoneal injection; IgY + DE: IgY-embedded double emulsion by gavage + LPS intraperitoneal injection.

**Figure 5 nutrients-16-03361-f005:**
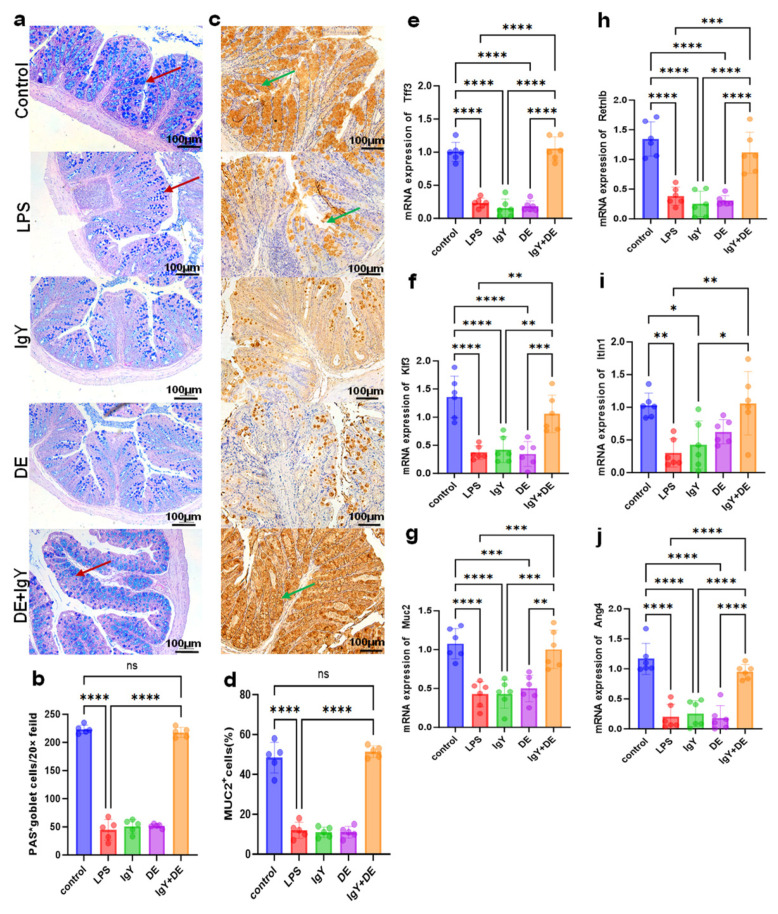
IgY + DE alleviates LPS-induced colitis by improving the mucosal barrier. (**a**) AB-PAS staining,100 μm. (Red arrows indicate goblet cells) (**b**) Enumeration of goblet cells (*n* = 5). (**c**) Immunohistochemistry staining of Muc2 in intestinal tissue sections. Bars: 100 μm, (Green arrows indicate Muc2 expression.) (**d**) %Area of Muc2 in the intestinal tissue from each treatment group (*n* = 5; 10 microscopic images were obtained for each treatment group). (**e**) Expression of *Tff3* was measured by real-time qPCR in mice (*n* = 5 per group). (**f**) Expression of *Klf3* was measured by real-time qPCR in mice (*n* = 5 per group). (**g**) Expression of *Muc2* was measured by real-time qPCR in mice (*n* = 5 per group). (**h**) Expression of *Retnlb* was measured by real-time qPCR in mice (*n* = 5 per group). (**i**) Expression of *Itln1* was measured by real-time qPCR in mice (*n* = 5 per group). (**j**) Expression of *Ang4* was measured by real-time qPCR in mice (*n* = 5 per group). Data were analyzed by one-way ANOVA: * *p* < 0.05, ** *p* < 0.01, *** *p* < 0.001, **** *p* < 0.0001. Control: physiological saline gavage + intraperitoneal injection of physiological saline; LPS: physiological saline gavage + LPS intraperitoneal injection; IgY: IgY gavage + LPS intraperitoneal injection; DE: double emulsion gavage + LPS intraperitoneal injection; IgY + DE: IgY-embedded double emulsion by gavage + LPS intraperitoneal injection.

**Figure 6 nutrients-16-03361-f006:**
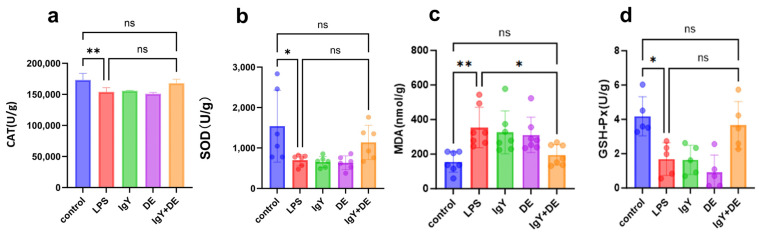
IgY + DE alleviates LPS-induced oxidative stress in the colon. (**a**) CAT expression in the colon (*n* = 6). (**b**) SOD expression in the colon (*n* = 6). (**c**) MDA expression in the colon (*n* = 6). (**d**) GSH-Px expression in the colon (*n* = 6). Data were analyzed by one-way ANOVA followed by Tukey’s post hoc test: * *p* < 0.05, ** *p* < 0.01. ns: represents no significance between the values. Control: physiological saline gavage + intraperitoneal injection of physiological saline; LPS: physiological saline gavage + LPS intraperitoneal injection; IgY: IgY gavage + LPS intraperitoneal injection; DE: double emulsion gavage + LPS intraperitoneal injection; IgY + DE: IgY-embedded double emulsion by gavage + LPS intraperitoneal injection.

**Figure 7 nutrients-16-03361-f007:**
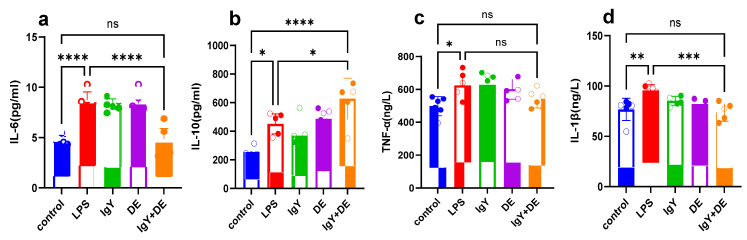
IgY + DE reduces the colonic inflammatory response caused by LPS. (**a**) Expression of IL-6 inflammatory factor in the colon (*n* = 6). (**b**) Expression of IL-10 inflammatory factor in the colon (*n* = 6). (**c**) Expression of TNF-α inflammatory factor in the colon (*n* = 6). (**d**) Expression of IL-1β inflammatory factor in the colon (*n* = 6). Data were analyzed by one-way ANOVA: * *p* < 0.05, ** *p* < 0.01, *** *p* < 0.001, **** *p* < 0.0001. ns: represents no significance between the values. Control: physiological saline gavage + intraperitoneal injection of physiological saline; LPS: physiological saline gavage + LPS intraperitoneal injection; IgY: IgY gavage + LPS intraperitoneal injection; DE: double emulsion gavage + LPS intraperitoneal injection; IgY + DE: IgY-embedded double emulsion by gavage + LPS intraperitoneal injection.

**Table 1 nutrients-16-03361-t001:** Primers for real-time PCR.

Gene Product	Primer Sequence (59–39)		
	Forward	Reverse	Source
*Muc2*	AGGGCTCGGAACTCCAGAAA	CCAGGGAATCGGTAGACATCG	NM_023566.4
*Tff3*	TTGCTGGGTCCTCTGGGATAG	TACACTGCTCCGATGTGACAG	NM_011575.2
*Klf3*	AAGCCCAACAAATATGGGGT	GGACGGGAACTTCAGAGAGG	XM_006503751.5
*Itln1*	TGACAATGGTCCAGCATTACC	ACGGGGTTACCTTCTGGGA	XM_029475723.1
*Retnlb*	AAGCCTACACTGTGTTTCCTTT	GCTTCCTTGATCCTTTGATCCAC	XM_021185515.1
*Ang4*	GGTTGTGATTCCTCCAACTCTG	CTGAAGTTTTCTCCATAAGGGCT	XM_021154346.1

## Data Availability

Data is contained within the article.

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
