# Peer review of "Protective Effect of IgY Embedded in W/O/W Emulsion on LPS Enteritis-Induced Colonic Injury in Mice"

_nutrients, 2024, doi:10.3390/nu16193361_

Round 1

Reviewer 1 Report

Comments and Suggestions for Authors

In the current work, the authors have developed double emulsion formulation of IgY and tested its efficacy against LPS 2 enteritis induced colonic injury.  The study is well executed.  However, the manuscript needs revision based on comments as under:

1.       Could you mention clearly the dose of IgY used in the formulation before administration.  Was dose escalation study performed before?  How did we decide on the selected dose?

2.       Double emulsion formulation has long term stability challenge upon storage.  Do we have long term stability data for good performing candidates?  Could you include in the manuscript how will you scale up the formulation for preclinical evaluation in higher primates?  What alternate formulation strategy you will approach is the current double emulsion formulation does not work for stability concerns?  Please include these responses in the manuscript with recent references.

3.       How did we quantify encapsulation efficiency?  Did we remove unencapsulated drug from the formulation and how did we achieve that?

4.       Throughout the manuscript, please keep degree sign consistent with respect to space between number and the degree symbol.

5.       What was the grade and source of PC used?  WPI is not approved by regulatory agencies for drug development.  Why did we select WPI for formulation development? How are we planning to sterilize these formulations given its higher particle size distribution. Please include your alternative approach with relevant references.

6.       Please include information related to a marketed formulation with efficacy against enteritis induced colonic injury as a working control to compare against the IgY encapsulated double emulsion efficacy in future preclinical efficacy experiments.  Thank you.

Comments on the Quality of English Language

Minor formatting and sentence completion errors noted on lines 30-31, 62, 137-146 (please re-write the entire paragraph with respect to manuscript writing and not as a protocol).

Author Response

1. Could you mention clearly the dose of lgY used in the formulation before administration. Wasdose escalation study performed before? How did we decide on the selected dose?

Response: Thank you very much for your question. The dose of IgY has been supplemented in 2.1. Construction of IgY embedded double emulsion. the dose of IgY was determined with reference to the dose of IgY in the previous study. The corresponding literature is also supplemented in the main text.

Location: page 2, lines 77-80.

2. Double emulsion formulation has long term stability challenge upon storage. Do we have longterm stability data for good performing candidates? Could you include in the manuscript how willyou scale up the formulation for preclinical evaluation in higher primates? What alternate formulation strategy you will approach is the current double emulsion formulation does not work for stability concerns? Please include these responses in the manuscript with recent references

Response: Thank you very much for your suggestion. Unfortunately, W/O/W emulsions still have the limitation of being difficult to store for long periods of time due to their double emulsion structure. In this study, it was found that the T80, SC and PC double emulsions showed varying degrees of water-oil delamination on the third day of storage at 4°C. Even the most stable WPI group double emulsions were found to be more stable than the WPI group double emulsions in the third day of storage. Even the most stable WPI group of double emulsions showed significant oil-water delamination on day 5. A previous study showed the same result, with a clear interface between the double emulsion groups (serum and cream layers) after 7 days of storage. We have supplemented the text with a detailed description of some of the subsequent measures that can be taken to address this limitation. Corresponding references are also added to the text.

Location: page 16, line number 391-418.

3.How did we quantify encapsulation efficiency? Did we remove unencapsulated drug from the formulation and how did we achieve that?

Response: Thank you very much for your question. We can control the encapsulation and release efficiency of double emulsion microcapsules by controlling the shear strength during the construction of W/O/W emulsions. In this study, we controlled the shear strength during the preparation of double emulsion to ensure the encapsulation efficiency of W/O/W. The corresponding manipulation is described in 2.1. Construction of IgY embedded double emulsion. It has also been studied to ensure the encapsulation and release efficiency by adding some emulsifiers. These have been added in the discussion section.

In this study, the prepared embedded IgY double emulsion as a dietary supplement needs to pass through gastric fluid to reach the intestine for its action, and according to the results of the study (Fig. 3b), the unembedded IgY was inactivated as soon as it entered the simulated gastric fluid. Therefore, the IgY that can successfully reach the intestine is stabilized by the double emulsion in the inner aqueous phase.

Location: page 16, line number 403-406, 419-430. page 2, line number 79-91.

4.Throughout the manuscript, please keep degree sign consistent with respect to space between 4number and the degree symbol.

Response: Thank you for your suggestion. The space between the degree symbol and the number 4 has been corrected throughout the text.

Location: page 4, line number 164, 187.

  1. What was the grade and source of PC used? WPl is not approved by regulatory agencies fordrua development, Why did we select WPl for formulation development? How are we planning tosterilize these formulations given its higher particle size distribution. Please include youralternative approach with relevant references

Response: Thank you for your comment. PC is an analytical standard grade soy lecithin. In order to prepare stable double emulsion-embedded IgY as a dietary supplement, natural macromolecule WPI was used as an emulsifier. Earlier studies have found that macromolecular emulsifiers (WPI) do not diffuse between interfaces, which is beneficial for the stability of W/O/W emulsions. Considering food applications, natural emulsifiers such as proteins (WPI) have the advantages of being biodegradable, biocompatible and non-toxic. It has been reported that the stability of W/O/W double emulsions can be improved if a low concentration of WPI is added to the internal aqueous phase. 

It has been shown that the addition of a small amount of non-toxic mustard oil or polymer-encapsulated gaseous chlorine dioxide (ClO2) in the construction of W/O/W emulsions can reach the goal of inhibiting the growth of bacteria and prolonging the stability of W/O/W emulsions.

Location: page 15, line number 353-361. page 16, line number 415-418.

  1. Please include information related to a marketed formulation with efficacy against enteritisinduced colonic injury as a working control to compare against the lgY encapsulated double emulsion efficacy in future preclinical efficacy experiments. Thank you.

Response: Thank you for your advice. The available commercially available agents that have shown efficacy against colonic injury due to enteritis are fluoroquinolones. A comparison of fluoroquinolones with IgY has been added to the main article.

Location: page 18-19, line number 527-540.

  1. Minor formatting and sentence completion errors noted on lines 30-31.62.137-146 (please re.the entire paragraph with respect to manuscript writing and not as a protocol).

Response: Thank you for your suggestion. The corresponding part has been modified in the original text.

Location: page 1, line number 30-31. page 2, line number 64-65 page 4, line number 168-183.

  1. IgY characteristics and quality

Response: Thank you for your question.The structure and quality of IgY is supplemented in the main text.

Location: page 1, line number 42-44.

  1. experiment is described inaccurantely

Response: Thank you very much for your suggestion, the subheading has been changed to 2.3.2. Tolerance test against artificial intestine juice

Location: page 3, line number 122.

  1. Details are missing

Response: Thank you for your advice. It has been added in detail in the main text.

Location: page 4, line number 158-160, 168-171, 174, 191-194.

  1. what do you mean by that?

Response: Thank you for your question. Emulsion rupture means that the emulsion shows severe water-oil partitioning (Fig. 1c)

Location: page 5, line number 213-214.

  1. which conditions. uplift stability ?

Response: Thank you for your question. during storage at 25°C. uplift stability modify to kinetic stability.

Location: page 5, line number 213-233.

  1. What are TSI values how are they measured refer Method Which changes are significant

Response: Thank you for your question. It has been added to the text. The Turbiscan Stability Index (TSI), calculated as the sum of all destabilization processes in the sample along the measurement cell, was quantified according to Equation (1).

TSI = ∑j |scanref(hj)− scani(hi)|

Where scanref and scani are the initial backscattering value and the backscattering value after 1 day of storage, respectively, hj is the given height in the measuring cell and TSI is the sum of all the scan differences from the bottom to the top of the tube.

Location: page 3, line number 100-113.

  1. How does the release occur, is there any model available? mechanism, release time quantity etc. Why is only one concentration tested?

Response: Thank you for your question. Consistent with previous results, the stability of the double emulsion remained essentially unchanged in the simulated oral and gastric fluid environments but decreased significantly upon entry into the simulated intestinal fluid. This suggests that the double emulsion is degraded by digestion in the intestinal fluid and IgY is released from the internal aqueous phase into the gut. It was found that some natural macromolecules (WPI, WPH) can form a thick and strong gel-like internal interface inside the double emulsion, and this structure is favorable for controlled release.

The dose of IgY has been supplemented in 2.1. Construction of IgY embedded double emulsion. the dose of IgY was determined with reference to the dose of IgY in the previous study. The corresponding literature is also supplemented in the main text.

Location: page 17, line number 442-449. page 2, lines 77-80.

Reviewer 2 Report

Comments and Suggestions for Authors

The manuscript entitled - Protective effect of IgY embedded in W/O/W emulsion on LPS 2

enteritis-induced colonic injury in mice – is dealing with an improved formulation against intestinal damage caused by continuously applied with LPS .

Basically, this investigation is interesting in respect to establish novel formulations with improved potency and compliance, which is well described in the introduction of the manuscript.

In the Material and Methods part, the applied methods and techniques are summarized. However, a more detailed description could substantially improve the manuscript quality and facilitate understanding.

The Result section contains detailed information about the W/O/W emulsions properties and short time stability. If available, data or estimates for longer stability could be included to assess the potential for clinical application. Specific comments are directly given in the manuscript.  Animals studies and efficacy data are performed scientifically correct and the results correspond to the expectations.

In the Discussion the authors discuss all results. Specific comments are  included in the text.

In conclusion the study is well done, however two main weaknesses are evident. The first is the incomplete method description and the second is that more detailed considerations of the release and the kinetics of the intestinal dynamic are missing. This could be done by alternative treatment strategies and more taken samples.

Comments on the Quality of English Language

Please check specific wording and typing errors

Author Response

  1. In the Material and Methods part, the applied methods and techniques are summarized. However, a more detailed description could substantially improve the manuscript quality andfacilitate understanding.

Response: Thank you for your suggestions. Specific experimental steps have been added to the Materials and Methods section.

Location: page 2, line number 77-96. page 3, line number 97-113. page 4, line number 158-171, 191-194.

  1. The Result section contains detailed information about the W/O/Wemulsions properties andshort time stability. lf available, data or estimates for longer stability could be included to assessthe potential for clinical application. Specific comments are directly qiven in the manuscript.

Response: Thank you very much for your suggestion. Unfortunately, W/O/W emulsions still have the limitation of being difficult to store for long periods of time due to their double emulsion structure. In this study, it was found that the T80, SC and PC double emulsions showed varying degrees of water-oil delamination on the third day of storage at 4°C. Even the most stable WPI group double emulsions were found to be more stable than the WPI group double emulsions in the third day of storage. Even the most stable WPI group of double emulsions showed significant oil-water delamination on day 5. A previous study showed the same result, with a clear interface between the double emulsion groups (serum and cream layers) after 7 days of storage. We have supplemented the text with a detailed description of some of the subsequent measures that can be taken to address this limitation.

Location: page 16, line number 391-418.

3.In conclusion the study is well done, however two main weaknesses are evident. The first is the incomplete method description and the second is that more detailed considerations of the releaseand the kinetics of the intestinal dynamic are missing, This could be done by alternative treatmenistrategies and more taken samples.

Response: Thank you for your advice. The corresponding materials and methods have been added inside the text; also some thoughts on the release of W/O/W emulsions in the gut have been added to the discussion (page 17, line number 442-449.).

Location: page 2, line number 77-96. page 3, line number 97-113. page 4, line number 158-171, 191-194. page 17, line number 442-449. 

  1. Please check specific wording and typing errors

Response: Thank you for your advice. The grammatical and wording errors in the article have been corrected.

Location: full text

Round 2

Reviewer 2 Report

Comments and Suggestions for Authors

Dear Authors,

Thank´s for your cooperation to substantially improve the quality of the manuscript. Now, I highly recommend to publish this manuscript.

Best reagards